# An inconsistency in aviation emissions between CMIP5 and CMIP6 and the implications for short-lived species and their radiative forcing

Robin N. Thor[1], Mariano Mertens[1], Sigrun Matthes[1], Mattia Righi[1], Johannes Hendricks[1], Sabine Brinkop[1], Phoebe Graf[1], Volker Grewe[1], Patrick Jöckel[1], and Steven Smith[2]

[1]Deutsches Zentrum für Luft- und Raumfahrt (DLR), Institut für Physik der Atmosphäre, Oberpfaffenhofen, Germany
[2]Joint Global Change Research Institute, Pacific Northwest National Laboratory, College Park, MD, USA

**Correspondence:** Robin N. Thor (robin.thor@dlr.de)

**Abstract.** We report on an inconsistency in the latitudinal distribution of aviation emissions between the data products of phases 5 and 6 of the Coupled Model Intercomparison Project (CMIP). Emissions in the CMIP6 data occur at higher latitudes than in the CMIP5 data for all scenarios, years, and emitted species. A comparative simulation with the chemistry-climate model EMAC reveals that the difference in nitrogen oxides emission distribution leads to reduced overall ozone changes due to aviation in the CMIP6 scenarios, because in those scenarios the distribution of emissions is partly shifted towards the chemically less active higher latitudes. The radiative forcing associated with aviation ozone is 7.6% higher and the decrease of methane lifetime is 5.7% larger for the year 2015 when using the CMIP5 latitudinal distribution of emissions compared to when using the CMIP6 distribution. We do not find a statistically significant difference in the radiative forcing associated with aviation aerosol emissions. In total, future studies investigating the effects of aviation emissions on ozone on climate should consider the inconsistency reported here.

## 1 Introduction

Emission data are a key contribution to the Coupled Model Intercomparison Project Phase 6 (CMIP6, Eyring et al., 2016). This framework provides both, historical emissions (Hoesly et al., 2018) and emissions for future scenarios (Riahi et al., 2017; Gidden et al., 2019). Apart from their usage within the CMIP itself, several studies have also used the aviation emissions provided within the framework of CMIP6 for other purposes, as they present an available dataset with future projections that are consistent with those of other sectors (e.g., Righi et al., 2021). The geographical and annual distribution of aviation emissions are identical throughout all historical and scenario data sets in CMIP6, leaving only the total annual emission amounts as variables that are different for each year and each scenario. According to the documentation (Hoesly et al., 2018), the geographical distribution of the CMIP6 aviation emissions is based on that of the CMIP5 aviation emissions (Lamarque et al., 2010), which in turn are derived from the Future Aviation Scenario Tool (FAST, Lee et al., 2005) for the European QUANTIFY project (Hoor et al., 2009), and is not affected by the regridding performed within CMIP6 (Feng et al., 2020).

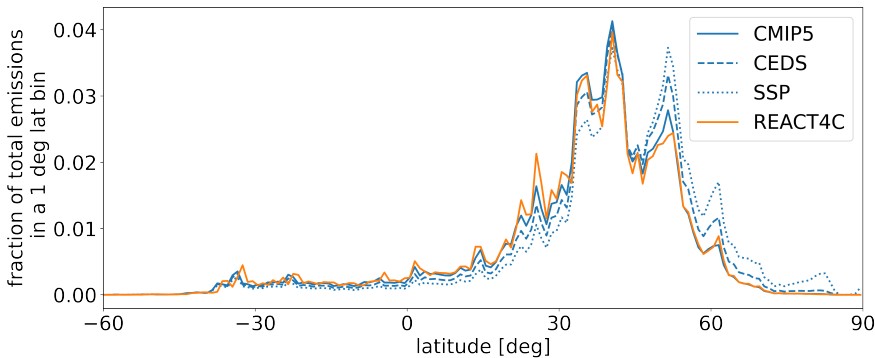

**Figure 1.** Fraction of total aviation emissions as a function of latitude. The solid blue line is based on the RCP 4.5 scenario for the year 2000 (CMIP5, Lamarque et al., 2010). The dashed blue line is based on historical emissions provided in the CEDS (CMIP6, Hoesly et al., 2018). The dotted blue line is based on the SSP2 4.5 scenario for the year 2015 (CMIP6, Fricko et al., 2017; Gidden et al., 2019). The orange line is based on the REACT4C inventory (Søvde et al., 2014).

Based on this information we would expect an identical geographical distribution of the aviation emissions in CMIP5 and CMIP6.

Here, we report on an inconsistency in the spatial pattern of aviation emissions between CMIP5 (Lamarque et al., 2010) and CMIP6 (Gidden et al., 2019; Feng et al., 2020). The latitudinal emission distribution differs by an approximate factor of $1.344\cos\phi$ for historic emissions provided in the Community Emissions Data System (CEDS, Hoesly et al., 2018) and by an approximate factor of $1.912\cos^2\phi$ for the Shared Socioeconomic Pathways (SSP) scenarios (Gidden et al., 2019), where $\phi$ is the latitude (Fig. 1). This difference is particularly noticeable in the North Polar region, where emissions are several times larger in the CMIP6 data sets, but in terms of total amount of emissions, the difference is largest in the regions from $\sim 50°$N to $\sim 65°$N and from $\sim 25°$N to $\sim 40°$N, where most emissions occur (see also Fig. 2). A comparison with an independent aviation emission inventory derived in the REACT4C project (Søvde et al., 2014) gives a very good match with the CMIP5 data set. The difference is observed for aviation emissions of nitrogen oxides ($NO_x$), black carbon (BC), and $CO_2$. Other emitted species (CO, $NH_3$, NMVOC, $SO_2$, organic carbon) have an identical geographic distribution to that of $NO_x$ and BC in CMIP6, but were not provided in the CMIP5 data. The factors by which the latitudinal emission distributions differ are independent of the year, as the geographical pattern of emissions is also constant over time in CMIP6. To obtain a aviation emissions that have the same total amount of emissions as in CMIP6, but exhibit approximately the same latitudinal distribution of emissions as the CMIP5 emissions, one has to multiply the CMIP6 CEDS historic emissions (until the year 2014) of all species by $1.344\cos\phi$ and the CMIP6 SSP scenario emissions (from the year 2015) of all species by $1.912\cos^2\phi$. The parameters 1.344 and 1.912 originate from the latitudinal distribution of aviation emissions and ensure that the total amount of emissions is not modified.

The aim of this paper is to investigate the impacts of the differences in the latitudinal distribution of emissions on aviation-induced ozone and aerosols and on their radiative forcing (RF). We do not consider emissions of $CO_2$, as it is a well-mixed

greenhouse gas with a long lifetime, implying that the spatial distribution of the emissions has a minor effect on the $CO_2$-induced climate effect. We also do not consider the potential differences in the contrail climate effect, because the CMIP data do not contain data on flight distance per area, which would be required for their computation (Bock and Burkhardt, 2019).

In Section 2, we introduce the used earth system model and simulation set-up and in Section 3, we present results on the aviation-induced atmospheric ozone concentration and aerosol distributions and differences in radiative fluxes.

## 2    Method

To investigate the effect of the difference in latitudinal distribution of aviation emissions on ozone, aerosols, and the related radiative forcings, we perform simulations with the chemistry-climate model ECHAM/MESSy Atmospheric Chemistry (EMAC),

using each of the geographical distributions, but identical total amounts of emissions. For the investigation of aviation-induced ozone changes, we perform a set of 5-year simulations, where the model is configured as a quasi-chemical transport model (QCTM) and uses a source apportionment (tagging) method. For the investigation of the aviation-induced aerosol effect, we perform a separate set of 13-year simulations, where the model is configured as a chemistry-climate model with nudged meteorology.

The EMAC model is a numerical chemistry and climate simulation system that includes sub-models describing tropospheric and middle atmosphere processes and their interaction with oceans, land and human influences (Jöckel et al., 2010). It uses the second version of the Modular Earth Submodel System (MESSy2) to link multi-institutional computer codes. The core atmospheric model is the 5th generation European Centre Hamburg general circulation model (ECHAM5, Roeckner et al., 2006). The physics subroutines of the original ECHAM code have been modularized and reimplemented as MESSy sub-models

and have been continuously been further developed. Only the spectral transform dynamical core, the flux-form semi-Lagrangian large scale advection scheme, and the nudging routines for Newtonian relaxation are remaining from ECHAM.

For the simulations of aviation-induced ozone changes in the present study, we applied EMAC (MESSy version 2.54.0.3) in the T42L90MA-resolution, i.e. with a spherical truncation of T42 (corresponding to a quadratic Gaussian grid of approx. 2.8 by 2.8 degrees in latitude and longitude) with 90 vertical hybrid pressure levels up to 0.01 hPa. The applied model setup

comprised RF calculations based on the sub-model RAD (Dietmüller et al., 2016) and the sub-model TAGGING (version 1.1, Grewe et al., 2017; Rieger et al., 2018) for the attribution of RF to emissions from the aviation sector (Mertens et al., 2018). The simulations use specified dynamics and the set-up is very similar to the one of the simulation RC1SD-base-10a described in detail by Jöckel et al. (2016). The gas phase mechanism is implemented using the sub-model Module Efficiently Calculating the Chemistry of the Atmosphere (MECCA, Sander et al., 2011) and incorporates the chemistry of ozone, methane and odd

nitrogen. Alkanes and alkenes are considered up to $C_4$, while the oxidation of $C_5H_8$ and some non-methane hydrocarbons (NMHCs) are described with the Mainz Isopren Mechanism version 1 (von Kuhlmann et al., 2004). Further, heterogeneous reactions in the stratosphere (sub-model MSBM, Jöckel et al., 2010) as well as aqueous phase chemistry and scavenging (SCAV, Tost et al., 2006) are included. Emissions of methane ($CH_4$) are not considered explicitly. Instead, $CH_4$ mixing ratios are relaxed towards observations using Newtonian relaxation with the sub-model TNUDGE (Kerkweg et al., 2006). Using the

sub-model LNOX, Lightning $NO_x$ is parameterised after Grewe et al. (2002) with global total emissions of $\sim 4.5$ Tg(N) a$^{-1}$,
which is within the range given by Schumann and Huntrieser (2007). Emissions of $NO_x$ from soil and biogenic $C_5H_8$ emissions
were calculated using the MESSy sub-model ONEMIS (Kerkweg et al., 2006), using parameterisations based on Yienger and
Levy (1995) for soil-$NO_x$ and Guenther et al. (1995) for biogenic $C_5H_8$.

    For one simulation we use the unaltered CMIP6 aviation emissions of the SSP2 4.5 scenario (Fricko et al., 2017) for the
year 2015, whereas for a second simulation we use the CMIP6 aviation emissions with their latitudinal distribution changed
to be equal to that of the CMIP5 emissions. Other simulation settings are identical. The presented results are obtained as a
5-year mean after a spin-up period of 6 months in QCTM mode (Deckert et al., 2011), where feedback between chemistry
and dynamics is suppressed, and using meteorology data reaching from 2013 to 2017 and specified dynamics by Newtonian
relaxation towards ECMWF ERA-Interim reanalysis data (Dee et al., 2011). For the spin-up period, we use meteorology data
from the second half of 2012. The simulations were initialized from a previous 1.5-year simulation including TAGGING for
the spin-up of the TAGGING tracers. This spin-up simulation itself was initialized from the long-term (since 1950) SC1SD-
base-01 simulation which is similar to the RC1SD-base-10a simulation (Jöckel et al., 2016).

    For the simulations of the aviation-induced aerosol effect, we used EMAC with the aerosol sub-model MADE3 (Modular
Aerosol Dynamics model for Europe, adapted for global applications, third generation; Kaiser et al., 2014, 2019) in the config-
uration described by Righi et al. (2020, 2023). With respect to the version adopted for the ozone changes, the EMAC-MADE3
setup for aerosol uses a lower vertical resolution with 41 layers, mostly covering the troposphere and the lower stratosphere,
and a simplified chemistry scheme, only including the reactions relevant for the aerosol processes. The aerosol simulations
cover a period of 13 years with nudged meteorology using the ECMWF ERA-Interim reanalysis data from 2006 to 2018, and
again using emissions for the year 2015. The QCTM mode and the tagging method cannot be applied for investigating the
aerosol effects, due to the role of the cloud feedback and the complexity of the liquid-phase chemistry for sulfate, respectively.
Hence, the statistical significance of the changes in the aerosol RF between the original and the corrected emission dataset is
evaluated using a paired sample $t$ test at the 95% confidence level.

## 3   Results

    Our analysis for the SSP scenarios shows that regional emission amounts from aviation differ substantially in the northern
mid-latitudes (Fig. 2). Emissions of $NO_x$ north of $45°N$ are 36.8% lower and emissions south of $45°N$ are 31.9% higher when
using the CMIP5 latitudinal distribution of emissions compared to when using the unaltered CMIP6 emissions. The mean
emission latitude shifts from $41.3°N$ for the CMIP6 latitudinal distribution to $34.3°N$ for the CMIP5 latitudinal distribution.

    The difference in the ozone distribution between the two QCTM simulations reflects the latitudinal difference in aviation
emissions (dashed and solid lines in Fig. 2, respectively). However, atmospheric dynamics and the larger chemical activity in
tropical latitudes lead to a southward shift of the ozone burden difference with respect to the emission difference. The increased
$NO_x$ emissions southwards of $45°N$ cause a positive ozone burden whose value (2.13 Tg) is larger than the absolute value of
the negative ozone burden caused by the decreased $NO_x$ emissions northwards of $45°N$ (-1.18 Tg). In total, using the CMIP5

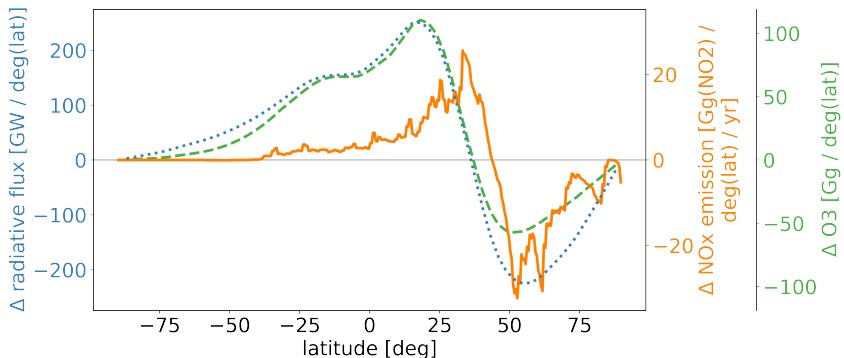

**Figure 2.** Differences (CMIP5 - CMIP6) of $NO_x$ emissions (solid orange line), $O_3$ burden (dashed green line), and radiative flux (dotted blue line) from two simulations with identical total amount of aviation emissions, but different latitudinal distributions, as a function of latitude.

latitudinal pattern of emissions increases the atmospheric ozone burden by 0.95 Tg, corresponding to an increase of 3.4% in the total ozone burden attributed to aviation.

We also compute the stratospherically adjusted radiative flux at the tropopause resulting from these differences in the ozone concentration distribution. The pattern of the radiative flux difference between the two simulations closely follows the pattern of the ozone burden difference, but the radiative flux decrease at high northern latitudes is more pronounced than the corresponding ozone decrease (Fig. 2). We show radiative flux instead of radiative forcing to keep all quantities in Fig. 2 independent of the area for better comparability. The radiative forcing attributed to aviation emissions is 30.82 mW m$^{-2}$ in the simulation

with unaltered CMIP6 emissions and 33.16 mW m$^{-2}$ in the simulation using the CMIP5 emission pattern, corresponding to a difference of 2.34 mW m$^{-2}$ or 7.6%. The difference in total RF between the two simulations is 2.08 mW m$^{-2}$. The total difference is smaller due to the non-linearity between nitrogen oxide emissions and ozone changes. Emissions from other sectors cause weaker radiative effects in a more polluted atmosphere, partly compensating for a larger aviation RF.

Transport emissions also influence the lifetime of methane, with aviation emissions generally leading to a lifetime decrease

(Mertens et al., 2022). In the simulation using the CMIP5 emission pattern, we found a 5.7% larger decrease of lifetime for aviation. All these changes are statistically significant because they are 5 to 6 times larger than their standard deviation over the 5-year simulation period. For context, aviation emissions have consistently increased over time, with a decadal increase ranging from 10-25%, depending on the time period (O'Rourke et al., 2021). This points to the importance of accurately quantifying not only the magnitude and spatial distribution of aviation emissions, but their changes over time.

Finally, we investigate the impact of the corrected emissions on aviation-induced changes in aerosol concentrations. This is shown in Fig. 3 for BC and reveals that adopting the CMIP5 latitudinal distribution of emissions results in a lower aviation-induced BC concentration compared to the unaltered CMIP6 inventory. Differences around 0.1–0.2 ng m$^{-3}$ are found throughout the troposphere at high latitudes ($> 40°N$). Even larger differences, up to 0.5 ng m$^{-3}$, can be seen at and below the typical cruise altitude ( ~200-300 hPa). Analogous differences are found for aviation-induced aerosol number concentration (not

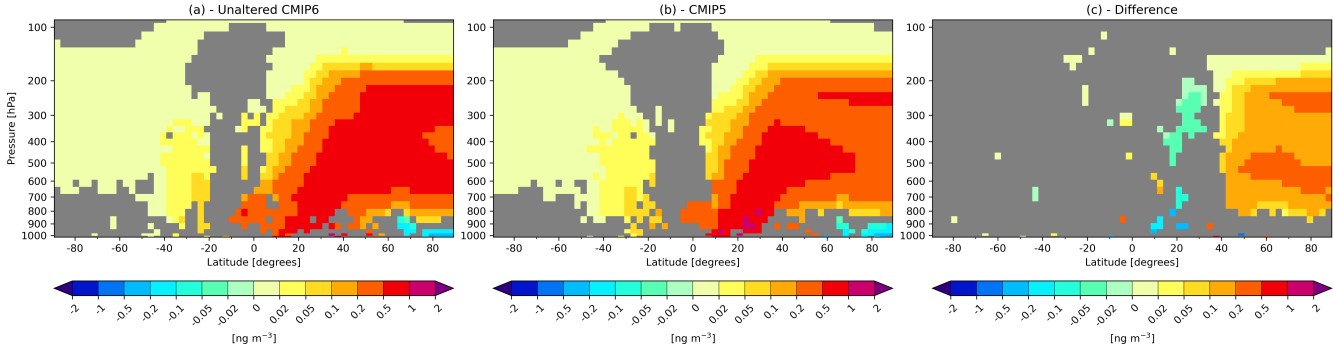

**Figure 3.** Zonally averaged aviation-induced changes in black carbon concentration simulated by the model using the unaltered CMIP6 (a) and CMIP5 (b) inventory, and their difference (c). Gray areas mark non-significant changes at the 95% confidence level.

shown). These changes can be relevant for the quantification of the impacts of aviation aerosol on climate, through their interactions with both warm clouds and cirrus (e.g., Gettelman et al., 2013; Righi et al., 2021) as well as ice surface albedo (Kang et al., 2020). The large variability of the climate system associated with aerosol-cloud interactions, however, hampers a robust quantification of the aerosol RF from aviation. Here, we quantify a RF of $-46.0 \, \text{mW m}^{-2}$ and $-54.7 \, \text{mW m}^{-2}$, for the simulations with the CMIP5 and unaltered CMIP6 emissions, respectively. This means that the climate impact of aviation due to aerosol is reduced (in absolute terms) by $8.7 \, \text{mW m}^{-2}$, i.e. 16%. This reduction is consistent with the aforementioned differences in aviation-induced particle concentrations, but its statistical significance is low (79.8%) for the reasons outlined above.

The effect of the difference in emissions on air quality is small. For example, the differences in the surface mixing ratio of ozone and nitrogen oxides between the two QCTM simulations are smaller than 0.5 % and 2.1 % at all locations, respectively. For nitrogen oxides, 98 % of the 2.8 by 2.8 degree grid cells in the model exhibit differences smaller than 0.2 %.

In this study, we are not considering the effect that a different latitudinal distribution of aviation soot emissions would have on contrail RF. According to Bock and Burkhardt (2019), a 50% reduction in soot emissions could lead to a 14% reduction in contrail RF. This implies a lower contrail RF north of $45°N$ and a higher contrail RF south of $45°N$ when using the CMIP5 latitudinal distribution of emissions compared to when using the unaltered CMIP6 emissions. The net effect may be a lower contrail RF due to the rarer occurrence of persistent contrails in tropical areas, but this effect is likely small due to the relatively small effect of soot emissions on contrail RF.

## 4 Conclusions

In summary, the inconsistency in the latitudinal distribution of aviation emissions between CMIP5 and CMIP6 leads to differences not only in the latitudinal distributions and regional emission amounts, but also in the total amounts of resulting ozone changes, methane lifetime changes, and RF attributed to aviation. The usage of the CMIP6 latitudinal distribution of emissions

leads to an overall lower climate effect of aviation emissions, even though the same total global amount of emissions was assumed in the simulations. The difference of 2.34 mW m$^{-2}$ reported in this study for the SSP2 4.5 scenario is small in the context of anthropogenic climate change, but constitutes 7.6% of the RF attributed to aviation ozone in our model. We therefore recommend that scholars studying the effects of aviation emissions on ozone and climate consider the inconsistency in the latitudinal distribution of aviation emissions reported here. We also investigated the effect of the inconsistency on aerosol RF, but could not detect a significant difference.

The impact of the inconsistency in the latitudinal distribution of aviation emissions on the RF and climate also depends on the background chemical composition of the atmosphere, which is a function of future global emission and pollution pathways. In a warmer and more polluted atmosphere, the chemical activity would be generally larger, particularly at high latitudes (Skowron et al., 2021). Therefore, the negative ozone burden change at Northern mid and high latitudes would likely be closer to the positive change at tropical and southern latitudes, leading to a smaller net relative effect of the inconsistency in terms of ozone burden and RF. The opposite would be expected for a less polluted atmosphere.

Furthermore, the results emphasize the importance of a correct and realistic geographic distribution of emissions when studying their effects on atmospheric composition and climate. Future aviation emission datasets should also consider temporal changes in the spatial distribution of emissions. No spatial changes over time were incorporated in either the CMIP5 or CMIP6 aviation datasets because such changes have not been estimated by the research community. The spatial distribution of aviation emissions have certainly changed over time, however (Quadros et al., 2022). For example, from 1990 to 2017 the share of estimated NO$_x$ emissions from flights originating in (roughly) the northern hemisphere (here Former Soviet Union, Europe, China, and North America) declined from 73% to 62%, implying a shift in aviation emissions away from the northern mid-latitudes (O'Rourke et al., 2021). Such shifts in the mean aviation emission latitude in the past and future have an impact on the climate effects of aviation NO$_x$ emissions, which many climate studies neglect.

We note in closing that the difference between CMIP5 and CEDS was found to be caused by an error in data pre-processing in CEDS and will be corrected in the next data release (Smith, 2022). This type of error can occur during conversion between masses and mixing ratios.

*Code and data availability.* The Modular Earth Submodel System (MESSy) is continuously further developed and applied by a consortium of institutions. The usage of MESSy and access to the source code is licensed to all affiliates of institutions which are members of the MESSy Consortium. Institutions can become a member of the MESSy Consortium by signing the MESSy Memorandum of Understanding. More information can be found on the MESSy Consortium website (http://www.messy-interface.org, last access: 10 October 2022). The simulations presented here have been performed with a release of MESSy based on version d2.54.0.3-pre2.55-02. All changes are available in the official release (version 2.55). The namelist setups used for the simulations and the scripts used for the creation of the figures are given in Thor (2022).

*Author contributions.* R. N. T. discovered the inconsistency. S. M. and V. G. conceptualized the study. M. M., R. N. T., and M. R. carried out the simulations. M. R., S. B., P. G., and P. J. prepared the modified CMIP6 input emission data for the model simulation to be consistent with the CMIP5 spatial emission pattern. S. B. calculated the methane lifetime. M. R. and J. H. calculated the aerosol RF. R. N. T. created all figures and wrote the manuscript with the help of all co-authors.

*Competing interests.* The authors declare that they have no conflict of interest.

*Acknowledgements.* This work used resources of the Deutsches Klimarechenzentrum (DKRZ) granted by its Scientific Steering Committee (WLA) under project ID bd0080. This research was funded by the European Union's Horizon 2020 research and innovation programme under grant agreement No. 101006742, project SENECA ((LTO) Noise and Emissions of Supersonic Aircraft). In addition, this study was supported by the DLR transport programme (projects Data and Model-based Solutions for the Transformation of Mobility - DATAMOST - and Transport and Climate - TraK) and by the DLR impulse project ELK (EmissionsLandKarte). The authors thank Helmut Ziereis for a thorough internal review. They also thank two anonymous referees for their careful reviews and I. Dedoussi for her comments.

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
