# Peer review of "An inconsistency in aviation emissions between CMIP5 and CMIP6 and the implications for short-lived species and their radiative forcing"

_Geoscientific Model Development, 2022_

## Author Comment (AC1)

**Reply to referees of the article "An inconsistency in aviation emissions between CMIP5 and CMIP6 and the implications for short-lived species and their radiative forcing"**

We thank the two anonymous referees for their comprehensive reviews. In the following, we comment on each of the raised points. Our answers are marked in italics.

**Anonymous referee 1**

In this study, the authors reveal an "inconsistency" in aviation emissions of the CMIP6 dataset. They then use the EMAC climate model to quantify the impact on simulated ozone radiative forcing and methane lifetime. That impact is small but, at a few percent, not negligible. The impact on aviation aerosol radiative forcing does not meet statistical significance criteria.

The study is short and to the point. It is well written, and the figures and analysis support the discussion well. In my comments, I mostly request clarification of the text in places. However, the aerosol discussion could benefit from being more detailed, and it would be useful to provide guidance on how to interpret past results based on the wrong emissions. For these reasons, I recommend minor revisions.

Main comments:

- Throughout the paper, the authors speak of an "inconsistency". That is diplomatic of them, but their analysis makes it clear that it is in fact a "mistake". They provide at least two lines of evidence that the zonal distribution of emissions was not supposed to be like that. Furthermore, I was surprised not to see any mention of the issue on the CEDS github website. There isn't even a new, corrected version available for download. Why not? There is a CEDS co-author, so they must be aware.

  - *Yes, the CEDS project is aware of this and this issue was flagged a couple months ago on the CEDS GitHub site. See: [https://github.com/JGCRI/CEDS/issues/45](https://github.com/JGCRI/CEDS/issues/45). There has not been a new CEDS data release since this issue was identified, but we have added a work-around noted in the GitHub issue ("for users that would like 100% compatibility with the CMIP5 era data, the CMIP5 3-D aircraft emission data can be scaled to match the global totals in the CEDS data."). The data will be corrected in the next data release. For now, we would like to maintain the wording "inconsistency" to reflect the larger point made in the conclusions that changes in the spatial distribution of aviation emissions can have noticeable impacts on model results, which calls for improved data that is both up to date and incorporates changes over time.*

- The paper quantifies the impact of the "inconsistency" for the year 2015, finding that it causes the ozone radiative forcing to be about 8% larger. Can that number be used as a rough correction factor for CMIP6-based studies that are already published? How does it vary in time?

  - *It is important to notice that only the ozone RF associated with aviation is 8% larger. The relative impact on the total ozone RF is much smaller. As mentioned in the conclusion, we think that the correction has important implications for studies investigating the aviation climate effect, but is rather not so important for studies investigating climate change on a broader scale.*

  - *The figure of 8 % can probably be used as a rough correction factor, but we recommend using corrected emissions for future studies. The impact of the emission correction*

*depends on the treatment of atmospheric dynamics and chemistry in each individual model, and therefore the effect may be smaller or larger than 8 % in any individual model study. The correction factor does not vary in time, except between the historic emissions and future emission scenarios, as described in the introduction of the paper.*

- ○ *We have added a paragraph in the conclusions section that explains the expected effects of a different background, as would be expected in future scenarios. We consider a detailed quantification of the dependence of the inconsistency's effect on the background atmospheric chemistry, which could be achieved by performing additional sets of simulations, out of the scope of this paper.*

- The impact on aerosol radiative forcing is discussed in a 5-line paragraph. That feels rushed. Granted, the impact on radiative forcing is affected by the large interannual variability in cloudiness. But can't the authors say something about the impact on aviation aerosol burden or residence time? The northward shift in emissions would certainly affect those.
  - ○ *The black carbon aerosol burden and aerosol number concentration would be lower in a scenario with the CMIP5 latitudinal distribution of emissions. We included a figure in the paper to show this difference and added some corresponding explanations in the text.*

Other comments:
- Line 6: Why single out ozone and leave out the result on methane lifetime?
  - ○ *We added the result on the methane lifetime to the abstract.*
- Line 9: The abstract also lacks a concluding sentence. What are the implications for aviation radiative forcing derived from CMIP6 emissions?
  - ○ *We added a concluding sentence to the abstract.*
- Lines 12-13: I suppose that statement tries to make the point that CMIP6 emissions are used beyond CMIP6. Perhaps then clarify that those studies used CMIP6 emissions despite not contributing to the CMIP6 database itself.
  - ○ *We changed the formulation of the sentence to better express that the mentioned studies make use of the CMIP6 outside of its original purpose.*
- Line 21: "differs by" is vague. I suggest clearly saying that CMIP6 distributions need to be multiplied by the given latitude-dependent factors to fix the problem discussed here. It would also be useful to elaborate on the coefficients. I suppose the 1.344 and 1.912 factors are related to the shape of the emission distribution along latitudes. Is it some kind of emission-weighted mean cosine (or cosine squared) of latitude? Finally, it would be useful to repeat here that the correction is time-independent because emission patterns do not vary with time in those datasets.
  - ○ *You are correct that the factors are an emission-weighted mean cosine (squared) of the latitude. We added two sentences to explicitly describe this and how to apply the correction.*
- Lines 29-30: I suggest spelling out the implication that CMIP6 emissions for those species also need correcting for the same factors as above.
  - ○ *We have also formulated this more explicitly in the sentences mentioned in the previous comment.*
- Lines 34-35: Could cite Bock and Burkhardt (2019), for example, https://doi.org/10.5194/acp-19-8163-2019 in support of that statement.
  - ○ *We added the citation.*
- Line 70: But again using emission data for 2015?
  - ○ *Yes, we clarified this in the text.*
- Caption of Figure 2: The lines have different styles, but also different colours, which should be mentioned.
  - ○ *We added the colors to the figure caption.*
- I am missing a concluding statement, here. Something saying that the shift in mean emission latitude probably had a small impact on the RF of aviation NOx emissions, an impact that is often neglected in climate studies. In addition, it would be worth noting that emissions will

probably shift again in the future, and that it would be good to have emission scenarios that allow the study of the climate impacts of that shift.

- o *We added a concluding sentence.*

Anonymous referee 2

I have read the paper entitled "An inconsistency in aviation emissions between CMIP5 and CMIP6 and the implications for short-lived species and their radiative forcing" by Thor et al.

It covers unstated differences in the spatial distribution of emissions between the CMIP5 and CMIP6 inventories and the implications for short-lived climate forcing and subsequent impacts.

This paper represents an important novel contribution to the scientific community, and I recommend it for publication in the journal Geoscientific Model Development.

The paper is well-written, and the methods and results are well-described. As such, I have no major comments, but please find a few specific comments below.

**General comments:**

The biggest question that I have is why these emissions inventories are different in the first place. Is there a likely calculation error performed by the emissions inventory community that would have led to this inconsistency? If there is a methodological error, it might be important to highlight some of the conversions that could have caused this inconsistency.

- *This error occurred due to an incorrect summation of vertical emissions when generating the normalized spatial proxy. As mentioned above, this has been corrected and the corrected data will be included in the next CEDS data release. We are not able to investigate the reason why the future emission data differs, but a workaround has been noted here: ([https://github.com/JGCRI/CEDS/issues/45](https://github.com/JGCRI/CEDS/issues/45).) We have added a paragraph at the very end of the paper that mentions this and reflects the current status of investigations.*

In addition, also show if this inconsistency changes with the CMIP emissions year.

- *We added a sentence to indicate that the inconsistency, just like the geographical distribution of emissions themselves, is constant over time*

Importantly, if the inventory is different for aviation, there might be other sectors for which the same distributional inconsistency exists.

- *The reported inconsistency is likely related to the specific processing required for international aviation emissions, whereas for other sectors emissions are reported and processed on national levels. We could not find any similar inconsistency in other sectors.*

A comment on what might have caused this inconsistency, and a comment in the conclusion of this paper indicating whether this inconsistency is found for other sectors would greatly help the scientific community to fix these types of differences in future inventories.

The paper shows that the difference is the largest for polar emissions (page 2, line 24). As a result, an additional difference in radiative impact may occur due to differences in the impact of black carbon on ice pack darkening. Since this is not mentioned, I assume it is not included in this study. If it is evaluated, please comment on it, or if not, please include a comment.

- *The set of aerosol simulations contain black carbon scavenging and the resulting albedo changes, but these outputs were not saved. The QCTM have identical aerosol representations and are therefore not affected. Therefore, we are unfortunately unable to quantify the ice pack darkening or the RF caused by this effect. We have added a figure to show the difference in aviation-induced black carbon concentration changes between the two emission inventories and some additional discussion in the text.*

**Line-by-line**

Line 34-35: The impact on contrails might be substantial too and it would be good to mention the differences could be significant. For support consider referencing:

- Bock, L., & Burkhardt, U. (2019). Contrail cirrus radiative forcing for future air traffic. *Chem. Phys*, *19*(12), 8163–8174. [https://doi.org/10.5194/acp-19-8163-2019](https://doi.org/10.5194/acp-19-8163-2019)
- Lund, M. T., Aamaas, B., Berntsen, T. K., Bock, L., Burkhardt, U., Fuglestvedt, J. S., & Shine, K. P. (2017). Emission metrics for quantifying regional climate impacts of aviation. *Earth System Dynamics*, *8*(3), 547–563. [https://doi.org/10.5194/esd-8-547-2017](https://doi.org/10.5194/esd-8-547-2017)
  - *The contribution of aircraft soot emissions to contrail RF is indeed a factor that we have not considered, but would contribute a small amount to the total RF difference caused by*

> *the inconsistency. We added a paragraph at the end of the results section where we qualitatively describe the expected effect. A more detailed analysis would be interesting, but as contrail formation was not investigated in the simulations performed for this study, we consider a more detailed quantification out of the scope of this manuscript.*

Methods section [Page 2 & 3]: This section describes the coupled climate chemistry model (line 39 to 49), but in the next paragraph it seems that the model is run with fixed meteorology using "quasi-chemical transport model" in the case of ozone, and for aerosols, "nudged meteorology … using reanalysis data". The methods section would be easier to follow by clarifying this earlier in the methods section, perhaps by stating earlier in the methods section how this model is applied (e.g. in line 41) before the description of the model continues.

> o *We added a summarizing paragraph to the beginning of the methods section which hopefully clarifies the differences between the two sets of simulations.*

Conclusion: I find this paper could elaborate on potential other implications of this difference in emissions distribution. For instance, would we also expect a difference in air quality impacts due to these differences?

> o *We added a paragraph at the end of the results section on the effect of the inconsistency on two indicators for air quality, the surface ozone mixing ratio and the surface nitrogen oxide mixing ratio. We conclude that the effect of the inconsistency on air quality is small.*